**Data Availability Statement:** Data remain available upon request due to patient confidentiality and

# Impact of mass administration of azithromycin as a preventive treatment on the prevalence and resistance of nasopharyngeal carriage of *Staphylococcus aureus*

**Soumeya Hema-Ouangraoua**[1]*, **Juliette Tranchot-Diallo**[1,2⦿], **Issaka Zongo**[3⦿], **Nongodo Firmin Kabore**[1⦿], **Frédéric Nikièma**[3], **Rakiswende Serge Yerbanga**[3], **Halidou Tinto**[3], **Daniel Chandramohan**[4], **Georges-Anicet Ouedraogo**[2], **Brian Greenwood**[4‡], **Jean-Bosco Ouedraogo**[3‡]

1 Centre MURAZ, Bobo-Dioulasso, Burkina Faso, 2 Université Nazi Boni, Bobo-Dioulasso, Burkina Faso, 3 Institut de Recherche en Sciences de la Santé (IRSS), Direction Régionale de l'Ouest (DRO), Bobo-Dioulasso, Burkina Faso, 4 London School of Hygiene & Tropical Medicine, London, United Kingdom

⦿ These authors contributed equally to this work.
‡ These authors also contributed equally to this work
* souangraoua.muraz@gmail.com

## Abstract

*Staphylococcus aureus* is a major cause of serious illness and death in children, indicating the need to monitor prevalent strains, particularly in the vulnerable pediatric population. Nasal carriage of *S. aureus* is important as carriers have an increased risk of serious illness due to systemic invasion by this pathogen and can transmit the infection. Recent studies have demonstrated the effectiveness of azithromycin in reducing the prevalence of naso-pharyngeal carrying of pneumococci, which are often implicated in respiratory infections in children. However, very few studies of the impact of azithromycin on staphylococci have been undertaken. During a clinical trial under taken in 2016, nasal swabs were collected from 778 children aged 3 to 59 months including 385 children who were swabbed before administration of azithromycin or placebo and 393 after administration of azithromycin or placebo. Azithromycin was given in a dose of 100 mg for three days, together with the anti-malarials sulfadoxine-pyrimethamine and amodiaquine, on four occasions at monthly inter-vals during the malaria transmission season. These samples were cultured for *S. aureus* as well as for the pneumococcus. The *S. aureus* isolates were tested for their susceptibility to azithromycin (15 g), penicillin (10 IU), and cefoxitine (30 g) (Oxoid Ltd). *S. aureus* was iso-lated from 13.77% (53/385) swabs before administration of azithromycin and from 20.10% (79/393) six months after administration (PR = 1.46 [1.06; 2.01], p = 0.020). Azithromycin resistance found in isolates of *S. aureus* did not differ significantly before and after interven-tion (26.42% [14/53] vs 16.46% [13/79], (PR = 0.62 [0.32; 1.23], p = 0.172). Penicillin resis-tance was very pronounced, 88.68% and 96.20% in pre-intervention and in post-intervention isolates respectively, but very little Methicillin Resistance (MRSA) was detected (2 cases before and 2 cases after intervention). Monitoring antibiotic resistance in *S. aureus*

deposited in the MURAZ computing center. The data remains available on request from the data manager Mr Diallo Ibrahima at cdcmuraz@gmail.com or dikydiallo@gmail.com. There are no barriers to accessing the data. You just need to make the access request to the data manager who will request the co-authors to agree before sharing.

**Funding:** This sub-study was ancillary to a large study (AZSMC), so we did not receive funding for it, but the AZSMC study was funded by the Joint Global Health Trials scheme; ClinicalTrials.gov number, NCT02211729.

**Competing interests:** The authors have declared that no competing interests exist.

and other bacteria is especially important in Burkina Faso due to unregulated consumption of antibiotics putting children and others at risk.

# Introduction

In 2018, despite progress over the past two decades, an estimated 5.3 million children under five-year old died, mainly from preventable or treatable causes such as birth complications, pneumonia, diarrhea, neonatal septicemia or malaria [1]. Sub-Saharan Africa remained the region of the world with the highest under-five mortality rate. The average under-five mortality rate was 78 deaths per 1,000 live births resulting in the death of one in 13 children before their fifth birthday [1].

In recent years, much attention has been paid to the potential of mass drug administration (MDA) of azithromycin (AZ) to reduce under-five mortality. Recent studies conducted in Ethiopia and the MORDOR trial in Malawi, Niger and Tanzania showed that MDA with AZ was associated with a significant reduction in child mortality, especially in infants [2,3]. However, a trial conducted in Burkina Faso and Mali in which AZ was given together with Seasonal Malarial Chemoprevention (SMC) drugs (sulfadoxine-pyrimethamine plus amodiaquine) did not detect any reduction in deaths or hospital admissions [4]. Several studies have documented the collateral effects of MDAs with AZ, including the emergence of antimicrobial resistance, which is a global public health concern. Studies on the acquisition of resistance after MDA have focused mainly on gut bacteria and *Streptococcus pneumoniae*. Studies in Tanzania, Nepal, and Gambia have shown no evidence of such resistance following a single treatment cycle [5–7]. Other studies have suggested that resistance can emerges after a single round mass treatment [8,9]. Recently, a study in Burkina Faso and Mali has shown the emergence of pneumococcal resistance after six semi-annual cycles of administration of azithromycin which persisted for a year after the last drug administration [10].

To date, little work has been done to assess the effect of AZ MDAs on other bacterial pathogens. A number of recent studies have looked at the gut resistome post azithromycin MDA uch as placebo-controlled trials reported by Doan et al. [11,12] have shown rates of macrolide resistance after multiple administrations of azithromycin that are approximately four times higher in cases than in controls. The main bacteria causing invasive diseases in children were, until fairly recently, *S. pneumoniae* and *Haemophilus influenzae*. However, following the widespread use of conjugate vaccines against *H. influenzae* type b (Hib) and *S. pneumoniae*, Hib infections have decreased significantly [13,14] and pneumococcal vaccine serotype infections are decreasing also [15]. The decrease in the incidence of the disease associated with these two pathogens has resulted in *S. aureus* becoming a relatively more common cause of invasive bacterial diseases than in the past [4].

In sub-Saharan Africa, *S. aureus* bacteremia is a common cause of invasive bacterial disease in children. Hospital studies have shown that *S. aureus* is the most common cause of invasive bacterial diseases in children under 5 years of age in Gambia and Nigeria [16,17].

This ancillary study to the trial conducted in Burkina Faso and Mali [4] has examined the impact of azithromycin administration as a preventive treatment on the prevalence of *S. aureus* nasopharyngeal carriage and its resistance to azithromycin. The impact of AZ administration of carriage of *S. pneumoniae* in the same population has been reported previously [10].

## Study population and methods

### Population

This study is a sub-study of a recently conducted trial which investigated the impact of adding AZ to the antimalarial drugs used for Seasonal Malaria Chemoprevention (SMC) in Burkina Faso and Mali [4]. In summary, 19,200 children aged from 3 to 59 months were randomized to receive Sulfadoxine-Pyrimethamine (SP) and Amodiaquine (AQ) (Guilin Pharmaceutical, Shanghai, China) with Azithromycin (AZ) (Cipla, Mumbai, India) or placebo (P). A household census was conducted in June 2014, and children of either sex who were 3 to 59 months of age on August 1, 2014, were eligible for enrollment in the trial. The household census was repeated in May 2015 and in May 2016 to recruit additional eligible children and to detect any deaths that had been missed through the surveillance system. The children were selected from all over the Houndé region. Each year, children who were still younger than 60 months of age on August 1 remained in follow-up for the subsequent trial year, and children who had reached 5 years of age on or before July 31 exited the trial on that date. Randomization was performed according to household to avoid the potential effect of within household transmission of infection; all eligible children who shared a kitchen were assigned to the same trial group. To mask the trial-group assignments for the trial team and caregivers, a placebo for azithromycin of identical appearance was used. Infants aged 3 to 11 months received 250 mg/12.5 mg of SP and 75 mg of AQ on day 1 and 75 mg of AQ on days 2 and 3. In addition, they received 100 mg of AZ or a AZ placebo on days 1, 2 and 3. Children between the ages of 1 and 4 received double of these doses. The drug combination SP-AQ was provided by the pharmaceutical company, Guilin Pharmaceutical (Shanghai, China) and Azithromycin and the corresponding placebo were provided by the CIPLA laboratory (Mumbai, India). All treatment doses were administered by trial staff. Coverage with monthly treatments was high, with more than 80% of children receiving three or four treatment cycles each year. Deaths, hospital admissions and clinic attendances were recorded throughout the study period [4]. Cross-sectional surveys were conducted at the end of each malaria transmission season [4] (Fig 1).

### Nasopharyngeal sampling

Nasopharyngeal swabs were obtained in 2014, 2015, and 2016 in July before AZ administration and in December after administration each year (hereafter referred to as pre- and post- interventions samples). Only samples taken from pre- and post-2016 interventions were evaluated for nasopharyngeal carriage of *S. aureus*. A total of 385 and 393 children, randomly selected from the 10,636 children who participated in the trial in Burkina Faso provided samples before and after administration of AZ or its placebo, respectively.

Swabs were taken from the posterior wall of a child's nasopharynx using a calcium alginate swab (FLOQSwabs™ Copan/USA) and immediately transferred to cryotubes containing a milk-tryptone-glucose-glycerol (STGG) medium. These cryotubes were tagged and placed in a cooler with ice packs before being transferred to the laboratory within eight hours of sampling and they were then kept at -80 degrees Celsius until analysis.

### Laboratory methods

The nasopharyngeal samples were thawed at room temperature and 10 μl of each sample were inoculated on Chapman medium (Oxoid Ltd) and incubated for 24 hours at 37 degrees Celsius. Presumed colonies of *S. aureus*, indicated by the presence of a golden yellow pigment, were purified by cultivation on fresh agar containing 5% fresh sheep's blood and incubated under the same conditions. The coagulase agglutination test (Slidex Staph plus,

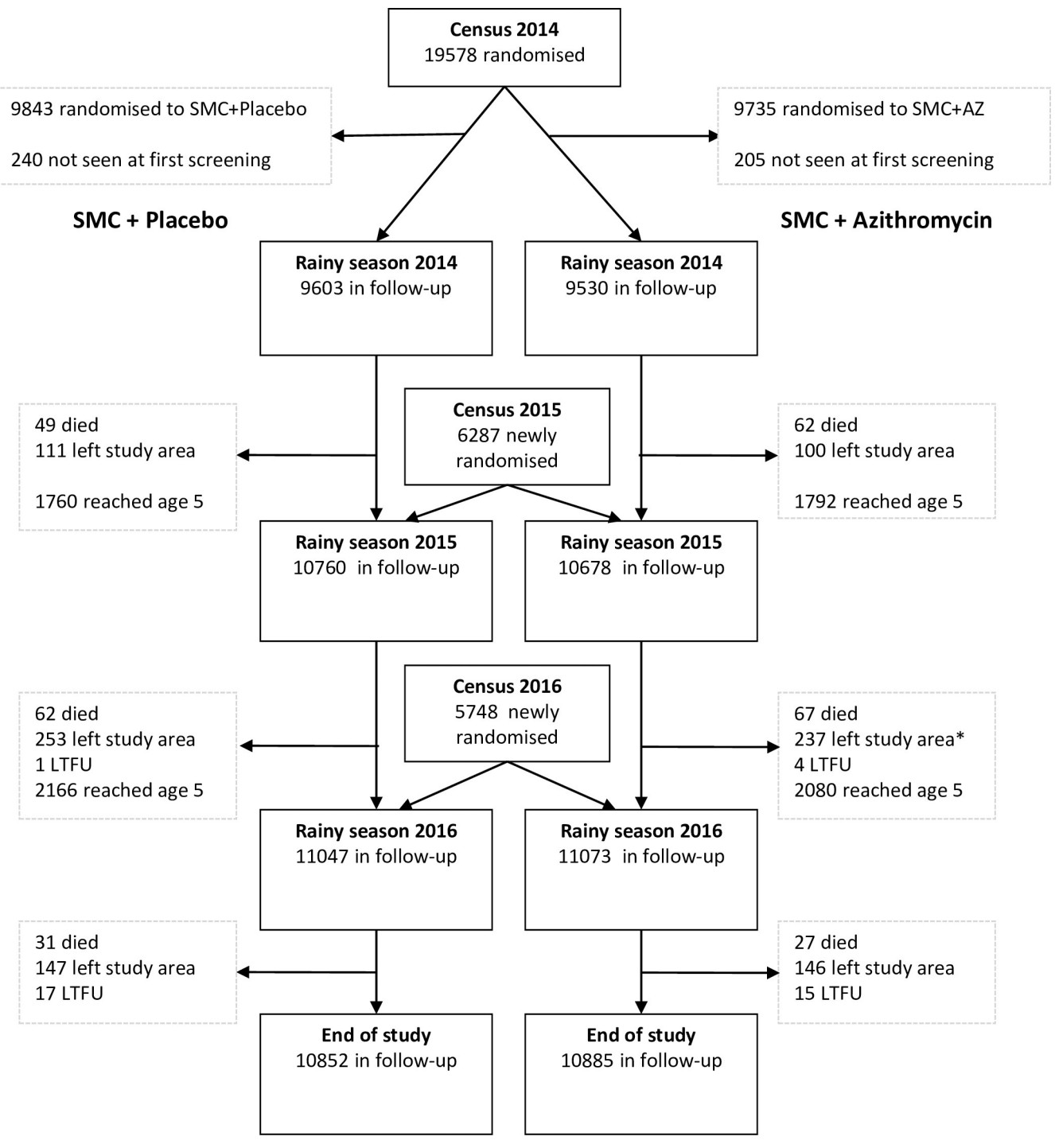

**Fig 1. Screening, randomization, and follow-up.**

BioMérieux®SA) was carried out on well-characterized colonies to confirm their identification as *S. aureus*. Standard 0.5 McFarland *S. aureus* colony suspensions were seeded in Mueller Hinton agar for the following antibiotic sensitivity tests: azithromycin (15 μg), penicillin (10 IU) and cefoxitin (30 μg) (Oxoid Ltd). Susceptibility results were interpreted in accordance

with Clinical and Laboratory Standards Institute [18]. Azithromycin resistance (AzmR) was defined by an inhibition diameter of 13 mm or less around the antibiotic disk, penicillin by an inhibition diameter of 28 mm or less, and that of cefoxitin by a diameter of 21 mm or less.

## Statistical considerations

The primary endpoint of the study was the prevalence of *S. aureus* carriage in intervention and control groups. A secondary endpoint was the overall prevalence of AZ-resistant *S. aureus* isolates in nasopharyngeal carriage. The evaluation criteria included an analysis of the sensitivity of staphylococci to other antibiotics. Demographic and biologic data are shown as proportions and compared by Pearson chi-square or Fisher's exact test. Prevalence ratio estimated by Poisson regression was used to compare prevalence. The significance threshold for all statistical tests was 0.05.

## Ethical considerations

The study was approved by the Ethics Committee of the London School of Hygiene and Tropical Medicine and the Ethics Committee of the Ministry of Health of Burkina Faso. The prospective study was recorded on Clinicaltrials.gov (NCT02211729). Written consent of the parents or guardians was obtained for the inclusion of a child in the overall trial and for inclusion in this sub-study.

# Results

## Study participants

A total of 385 children (193 in the Azithromycin group, and 192 in the placebo group) were swabbed in 2016 before the intervention, and 393 (192 in the Azithromycin group and 201 in the placebo group) were swabbed after the intervention. Demographic and biological data of trial participants are given in Table 1. Only 7.07% of study participants carried both pneumococci and staphylococci.

## Carriage with *S. aureus*

*S. aureus* carriage before and after intervention were shown in Table 2. There was no difference in *S. aureus* carriage between the AZ and placebo groups before intervention (p = 0.899) or after the intervention (p = 0.267).

## Frequency of azithromycin-resistant *S. aureus*

The overall frequency of *S. aureus* resistance to azithromycin (AzmR) was 26.42% ((14/ 53) [CI 95% 16.15; 40.08] and 16.46% (13/79) [CI 95% 9.72; 26.49], respectively before and after intervention. There was unable to find a significant difference in this relatively small sample.

**Table 1. Socio-demographic characteristics of the study population.**

| | | Pre-intervention 2016 (before) | | | Post intervention- 2016 (after) | | |
|---|---|---|---|---|---|---|---|
| | | Azithromycin N = 193 | Placebo N = 192 | p-value* | Azithromycin N = 192 | Placebo N = 201 | p-value* |
| Age (years) group, n (%) | <1 | 25 (13) | 32 (16.67) | 0.283 | 17 (8.85) | 16 (7.96) | 0.60 |
| | 1–2 | 90 (46.63) | 75 (39.06) | | 93 (48.44) | 89 (44.28) | |
| | 3–5 | 78 (40.41) | 85 (44.27) | | 82 (42.71) | 96 (47.76) | |
| Males, n (%) | | 87 (45.08) | 88 (45.83) | 0.882 | 89 (46.35) | 99 (49.25) | 0.57 |

*p value for the comparison between AZ and placebo groups. Pearson chi-square test was used for these comparisons.

**Table 2. Microbiological characteristics of the study population.**

|  | Pre-2016 (before) | | | Post- 2016 (after) | | |
|---|---|---|---|---|---|---|
|  | Azithromycin N = 193 | Placebo N = 192 | P value | Azithromycin N = 192 | Placebo N = 201 | P value |
| Recent antibiotic use, n(%) | 13 (34.21) | 13 (40.63) | 0.580 | 8 (66.67) | 9 (56.25) | 0.705 |
| *S. pneumoniae (Sp)* carriage, n(%) | 95 (49.22) | 103 (53.65) | 0.442 | 84 (43.75) | 99 (49.25) | 0.311 |
| *Sp/AZ resistance*, n(%) | 9 (6.38) | 11 (10.7) | 0.761 | 22 (25.6) | 13 (13.1) | 0.034 |
| *S. aureus (S.a) carriage*, n(%) | 27 (13.99) | 26 (13.54) | 0.899 | 43 (22.40) | 36 (17.91) | 0.28 |
| *S.a/AZ resistance*, n(%) | 8 (29.63) | 6 (23.08) | 0.589 | 9 (20.93) | 4 (11.11) | 0.24 |
| Double carriage, n(%) | 10 (5.18) | 12 (6.25) | 0.651 | 17 (8.85) | 16 (7.96) | 0.75 |

*p value for the comparison between AZ and placebo groups. Pearson chi-square test was used for these comparisons.

There was also no difference in *S. aureus* resistance according to the arm of treatment (PR = 1.51 [0.74; 3.07], p = 0.259) (Fig 2A).

## Frequency of resistance to other antibiotics

The overall prevalence of penicillin resistance (both arms combined) was very high—88.68% [95% CI 76.72; 94.90] before and 96.20% [95% CI 88.70; 98.79] after the intervention (PR = 1.08 [0.97, 1.21], 1.21] p = 0.137). Resistance to penicillin did not vary according to the treatment arm (PR = 1.22 [0.98; 1.52], p = 0.076) (Fig 2B).

Only 4 cases of cefoxitin-resistant *S. aureus* (MRSA) were detected; two cases before the intervention both in the azithromycin arm and two after intervention, one in each arm.

## Discussion

*S. aureus* is a well-known pathogen whose resistance to most available antimicrobial agents is increasing alarmingly. In this study, a 13.77% nasopharyngeal carriage rate of *S. aureus* was found prior to the intervention, a similar prevalence of *S. aureus* carriage to that found in an Ethiopian study in 2017 that reported a prevalence of 13% (52/400) among pre-school children [19]. On the other hand, the rate found in this study was a little lower than that of a Ugandan child population in the same age group with a carriage rate of 19.4% [20] or in a Ghanaian population where it was 21% [21]. After mass administration of azithromycin in the community, the prevalence of *S. aureus* carriage increased up to 20.10% in contrast to the carriage rate of pneumococci in the same children which fell from 51.69% to 46.71% in 2016. Azithromycin resistance for *S. aureus* did not increase after administration of the antibiotic overall or in either arm (PR = 1.51 [0.74; 3.07], p = 0.259). In contrast this was not the case for pneumococcal resistance, where an increase in azithromycin resistance was seen in children in the same study and was more significant in children treated with azithromycin PR = 1,95 (1,05, 3,61) P = 0,034 [10]. However, an effect might have been seen had AZ administration continued for more than one year as was seen when resistance of pneumococci to AZ was followed for a longer period [10]. A high prevalence rate of resistance to penicillin was found with more than 90% of the strains produced penicillinase. This rate of resistance to *S. aureus* to penicillin is similar to that reported in most Africa studies [19–22]. However, the resistance to cefoxitin found in our study (4 cases) was lower estimates than in some other studies [22,23].

While it is one of the most common commensals in normal flora, *S. aureus* is a dangerous pathogen that has developed resistance to nearly every new antibiotic introduced in half a century. The plasticity of its genome gives the bacterium the ability to adapt to all environmental conditions, including acquiring antibiotic resistance genes and developing regulatory mechanisms to

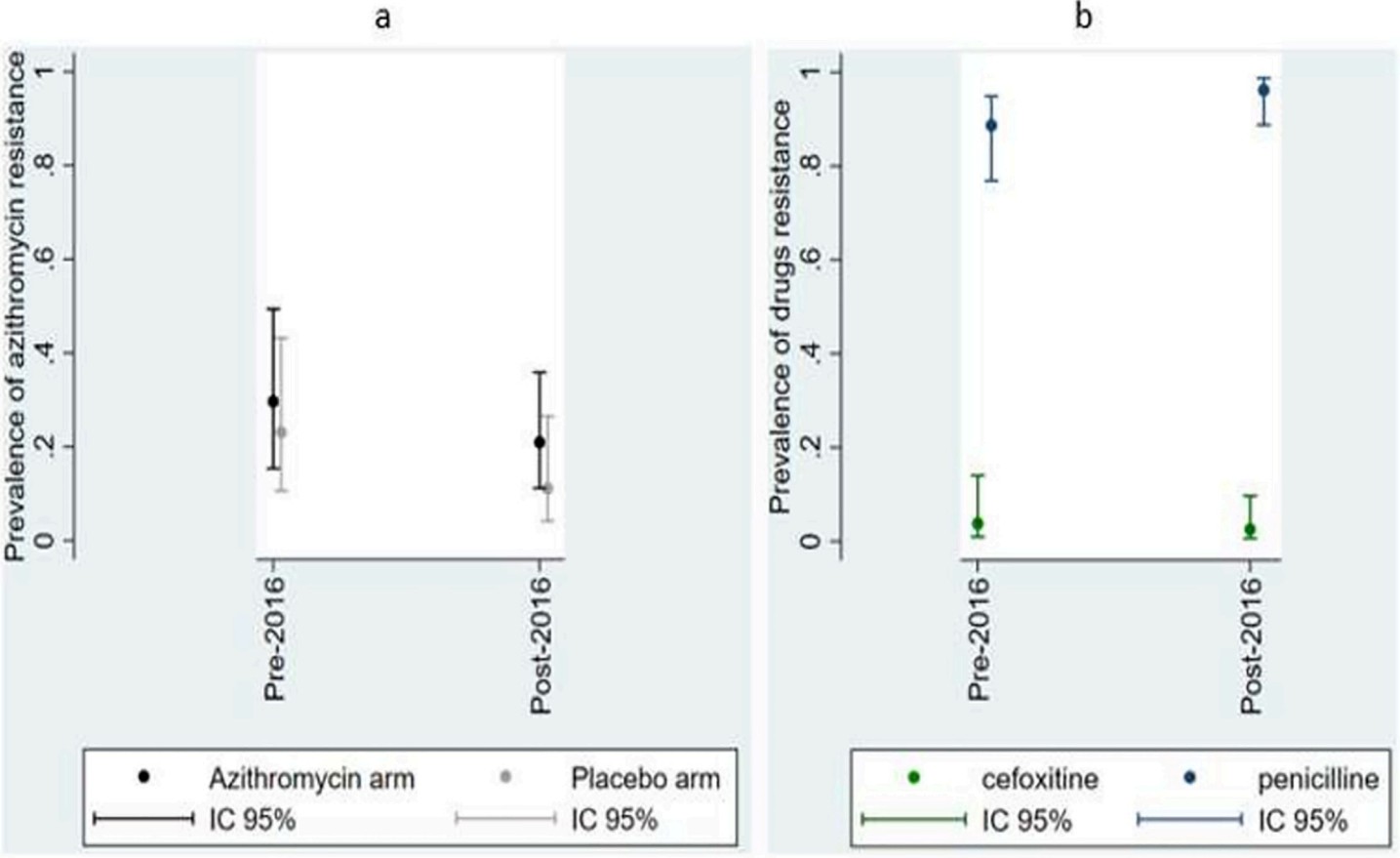

**Fig 2.** Prevalence of *S. aureus* resistance to antibiotic: a) Resistance to azithromycin by arm before and after intervention; b) resistance to penicillin and cefoxitin before and after administration of azithromycin.

adapt to increasing concentrations of antibiotics. Although MDA with azithromycin was not found to lead to an increase in resistance of *S. aureus* to this antibiotic in this study; The relatively wide CIs would suggest that further studies are necessary to better assess AMR.

## Supporting information

**S1 File.**
(DOC)

**S2 File.**
(PDF)

## Acknowledgments

The authors thank the Ministry of Health staff in Houndé district for their assistance: the lab technicians, data clerks, field workers and supervisors for data collection, and all the caretakers and children for their participation.

## Author Contributions

**Conceptualization:** Soumeya Hema-Ouangraoua, Daniel Chandramohan.

**Data curation:** Soumeya Hema-Ouangraoua, Nongodo Firmin Kabore, Frédéric Nikièma.

**Formal analysis:** Soumeya Hema-Ouangraoua, Nongodo Firmin Kabore.

**Investigation:** Soumeya Hema-Ouangraoua, Issaka Zongo, Georges-Anicet Ouedraogo.

**Methodology:** Soumeya Hema-Ouangraoua, Frédéric Nikièma, Daniel Chandramohan, Brian Greenwood.

**Resources:** Brian Greenwood.

**Supervision:** Frédéric Nikièma, Daniel Chandramohan.

**Validation:** Soumeya Hema-Ouangraoua, Issaka Zongo, Nongodo Firmin Kabore, Brian Greenwood.

**Visualization:** Soumeya Hema-Ouangraoua, Brian Greenwood.

**Writing – original draft:** Soumeya Hema-Ouangraoua, Brian Greenwood.

**Writing – review & editing:** Juliette Tranchot-Diallo, Issaka Zongo, Nongodo Firmin Kabore, Frédéric Nikièma, Rakiswende Serge Yerbanga, Halidou Tinto, Daniel Chandramohan, Georges-Anicet Ouedraogo, Brian Greenwood, Jean-Bosco Ouedraogo.

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
