## [Decision Letter · Decision Letter 0]

3 Mar 2021

PONE-D-20-33805

Impact of mass administration of azithromycin as a preventive treatment on the prevalence and resistance of nasopharyngeal carriage of Staphylococcus aureus

PLOS ONE

Dear Dr. Ouangraoua,

Thank you for submitting your manuscript to PLOS ONE. After careful consideration, we feel that it has merit but does not fully meet PLOS ONE’s publication criteria as it currently stands. Therefore, we invite you to submit a

We look forward to receiving your revised manuscript.

Kind regards,

Ray Borrow, Ph.D., FRCPath

Academic Editor

PLOS ONE

Journal Requirements:

2. Please confirm that you have included all items recommended in the CONSORT checklist including identifying the study as a randomized trial in the title.

3.We note that you have indicated that data from this study are available upon request. PLOS only allows data to be available upon request if there are legal or ethical restrictions on sharing data publicly. For information on unacceptable data access restrictions, please see http://journals.plos.org/plosone/s/data-availability#loc-unacceptable-data-access-restrictions.

4. Please amend the manuscript submission data (via Edit Submission) to include author Georges-Anicet Ouédraogo.

5. Please upload a copy of Figure 1, to which you refer in your text on page 8. If the figure is no longer to be included as part of the submission please remove all reference to it within the text.

6. Please ensure that you refer to Figure 2 in your text as, if accepted, production will need this reference to link the reader to the figure.

Reviewers' comments:

Reviewer's Responses to Questions

**Comments to the Author**

1. Is the manuscript technically sound, and do the data support the conclusions?

Reviewer #1: Yes

Reviewer #2: No

Reviewer #3: Yes

2. Has the statistical analysis been performed appropriately and rigorously? 

Reviewer #1: Yes

Reviewer #2: No

Reviewer #3: No

3. Have the authors made all data underlying the findings in their manuscript fully available?

Reviewer #1: Yes

Reviewer #2: No

Reviewer #3: No

4. Is the manuscript presented in an intelligible fashion and written in standard English?

Reviewer #1: Yes

Reviewer #2: Yes

Reviewer #3: Yes

5. Review Comments to the Author

Reviewer #1: In this study, the authors examined whether mass drug administration of azithromycin induces antimicrobial resistance to S. aureus in children in Burkina Faso when administered with anti-malarial meds. Prevalence of the infection was also estimated before and after intervention. Data for this study were collected as part of a larger trial of 19,200 children in Burkina Faso and Mali. This secondary analysis of data from that study parallels a prior study by the lead author on resistance to Streptococcus pneumoniae.

Comments:

• Only children from Burkina Faso were included in this study. Why were data on children from Mali excluded? Given that these data were derived from a published clinical trial which included roughly twice as many children, could the authors explain why only half the children were included? The study of resistance to S. pneumoniae included all children from both B.F. and Mali.

Introduction

• It seems odd that this is not identified as a sub-study of one of the studies that is referenced (ll.53-56). Although this is mentioned in the Study Population section, I think that this information should be clearly stated in the introduction (rather than referring to the parent trial as if it were a separate study).

Methods

• Why were swabs from the third year only of the study included? Again, the study on S. pneumoniae contained data from all three years. Could the authors explain why they chose to limit the data to the third year only?

• Although several studies have already been published from this trial, I still think it would be helpful to include some details about how the children were enrolled (in a dynamically stable cohort?) over the three-year period. In other areas as well, the paper would benefit from being a little more fleshed out with details, for those who have not read the other papers.

Results

• ll.136-37 I found this somewhat confusing as written. A clearer description would include the numbers shown in Table 1: “A total of 385 children (193 in the Azithromycin group, and 192 in the placebo group) were swabbed in 2016 before the intervention, and 393 (192 in the Azithromycin group and 201 in the placebo group) were swabbed after the intervention.” Not all children were administered AZ, so it would be more correct if the authors referred to pre- and post-intervention, rather than pre- and post-administration of AZ.

• As the authors stated that the primary objective of their study was to examine the prevalence of azithromycin-resistant S. aureus before and after the intervention, this section should precede the section on carriage with S.aureus (although that is what is stated in the Statistical Consideration section, in the introduction it is stated that the study “has examined the impact of azithromycin administration as a preventive treatment on the prevalence of S. aureus nasopharyngeal carriage and its resistance to azithromycin”). These two statements should be consistent with one another.

• The authors compare rates of S. aureus carriage in the treatment and intervention groups both before and after the intervention. They also present rates before and after within the two groups. This is confusing, given that the first set of results are presented in Table 1 and the second set are not. Perhaps a table that clearly presented both sets of results could be substituted. The same is true in their reporting of the prevalence of arithromycin-resistant S. aureus. Beautifully detailed papers are presented in the prior article on S. pneumoniae, but the table in this paper is very basic and uninformative.

While this study presents interesting and important data and is well-written, I feel like it suffers from comparison to the previous article by the same author on S. pneumoniae. It seems to have been less thoroughly investigated and much more briefly reported. I’m sure the author does not want to write the same paper twice, but I think the current paper would benefit from a more detailed exposition, and some explanation of why the authors chose to drastically cut the cohort and shorten the approach taken in the prior paper.

Reviewer #2: Interesting randomized controlled trial of staph resistance post azithromycin treatment. My concerns are mostly minor and easily addressable:

1. References 5-10. Your references highlight cross-sectional or longitudinal monitoring of AMR post azithromycin MDA. There are a large number of RCTs clearly showing selection of macrolide-resistant strains of pneumococcus after azithro MDA which were not referred to. If you’re not going to review that literature, consider just citing a recent systematic review (O’Brien Lancet ID, 2019).

2. Page 3, line 64. “To date, little work has been done … other bacterial pathogens.” A number of recent studies have looked at the gut resistome post azithromycin MDA. This is in a sense looking at AMR in all bacteria, so your sentence could be reworded. You could consider referring to them (starting w/ Doan, NEJM 2019 and 2020).

3. Line 102. While the nasopharynx may be ideal for pneumococcus, would the nares have been optimal for Staphylococcus aureus recovery?

4. False precision throughout the paper may mislead casual readers. Eg, Table 1. 25 of 193 is 12.95%? 13.0% or even 13% would be fine given the limited number of samples. Roughly an extra digit throughout the paper may mislead the reader. Similarly, P-values .602 could be .60.

5. Lines 147-150. “Carriage increased after AZ administration in both the AZ and placebo

6. groups with the increase being more marked and statistically significant in the AZ group (PR=1.46 [1.06; 2.01], p=0.020 than in the children who received placebo in whom the increase was not statistically significant (PR=1.21 [0.64; 2.30], p=0.558.” Please check the meaning of this sentence, I had difficulty parsing. The most relevant contrast would be the follow-up carriage comparing the children randomized to azithro- vs placebo. The longitudinal comparisons within an arm are not nearly as interesting, as carriage or resistance could change for any number of reasons other than your intervention.

7. Line 153. Consider rewording “comparable”. The CI is quite broad and includes values which many might not consider “comparable”, so better to just say unable to find a significant difference in this relatively small sample.

8. Lines 178-180. Consider highlighting the RCT comparison between arms rather than the longitudinal comparisons within an arm, which are not as informative.

9. Line 187. Consider rewording “significantly lower” to just “lower estimates”, as I’m not sure statistical significance could actually be tested in surveys with different methods in different countries at different times.

10. Line 197-199. Semicolon could be a comma? Your conclusion, that “it is essential that antibiotic resistance is monitored …” is certainly not warranted from the results of this study. In fact, you were unable to show an increase. You might say “may still be prudent to monitor”. Or you could say that the relatively wide CIs would suggest that further studies are necessary to better assess AMR. But I think the conclusion should reflect the study’s results, not opinion.

11. The Figure is completely redundant with the text. Just noting, fine to leave in if you feel it conveys information in a better way.

12. I understand that the azithromycin was the randomized intervention, but you were also giving multiple doses of the antibiotic sulfidoxine. You could consider mentioning that monitoring sulfa-resistance might also be of interest.

Reviewer #3: This is a fairly straightforward report of resistance in Staph aureus isolates from a randomized trial that assessed mass azithromycin distribution vs placebo in the setting of seasonal malaria chemoprevention. I have a few suggestions to clarify the reporting.

Line 82: I think this would be much clearer if phrased: “…children received SP/AQ for seasonal malaria chemoprevention, and were randomized to either azithromycin or placebo.” Or something like that. Because right now, it sounds like one group received SP/AQ plus azithro, and the other group got placebo.

Line 82: perhaps I missed it, but please provide some context about the study setting. How many villages contributed to the study? What types of villages? Is anything known about the antibiotic consumption in these villages or similar places of Burkina Faso? This is important for generalizability of the results.

Line 99: please provide some details about the random selection. The underlying trial randomized households to the treatments. Were swabs also performed on all kids from randomly selected households? Or a simple random sample? Were children all taken from a single village? How many villages? This is important because the topic is a transmissible infection, and both carriage and resistance will depend not only on the intervention but also on the interventions that siblings and neighbors get.

Line 136: this reads as if all 778 children received both swabs. But I don’t think this is the case? Consider clarifying the wording.

Line 152: the “prevalence” estimate would seem to use only those kids with Staph aureus carriage as the denominator. Is this actually a prevalence? Seems like it’s moreso a frequency among the children carrying Staph aureus? Also, it would be good to clarify for the reader what the denominator is (ie, all kids regardless of Staph aureus status, vs only those with positive Staph aureus results)

Line 155/160: if the estimates being compared are only from kids with a positive Staph aureus isolate, this means the clinical trial comparison would be conditioning on a post-randomization factor (ie whether the kid grew Staph aureus), which could induce bias. It would seem less biased to me to define resistance based on the entire random sample. At least for the RCT comparison.

Line 169: I always thought that the niche for Staph aureus was the nares. And yet here the nasopharynx was sampled. It might be nice to have a few sentences in the discussion about the site of sampling and whether nasopharyngeal carriage estimates would be similar to nares carriage estimates.

6. PLOS authors have the option to publish the peer review history of their article (what does this mean?). If published, this will include your full peer review and any attached files.

Reviewer #1: No

Reviewer #2: No

Reviewer #3: No

---

## [Author Response · Author response to Decision Letter 0]

15 Jul 2021

Dear Editorial Board,

Re: Revised manuscript Ref.: PONE-D-20-33805

Impact of mass administration of azithromycin as a preventive treatment on the prevalence and resistance of nasopharyngeal carriage of Staphylococcus aureus for publication in PLOS ONE

Revised manuscript titled 

Thank you for reviewing our manuscript. 

We appreciate the reviewers’ insightful comments and we have revised the manuscript accordingly as outlined in the attached responses. 

 This is corrected

2. Please confirm that you have included all items recommended in the CONSORT checklist including identifying the study as a randomized trial in the title.

 I confirm

3.We note that you have indicated that data from this study are available upon request. PLOS only allows data to be available upon request if there are legal or ethical restrictions on sharing data publicly. For information on unacceptable data access restrictions, please see http://journals.plos.org/plosone/s/data-availability#loc-unacceptable-data-access-restrictions.

the individual data from this study cannot be made public as there is no consent from the patients in this sense.

However, if a researcher wants to conduct research on the database, he or she submits a request to the research team, which will be submitted to the consortium to decide on the matter.

4. Please amend the manuscript submission data (via Edit Submission) to include author Georges-Anicet Ouédraogo

The author Georges-Anicet Ouédraogo has added

5. Review Comments to the Author

Reviewer #1: In this study, the authors examined whether mass drug administration of azithromycin induces antimicrobial resistance to S. aureus in children in Burkina Faso when administered with anti-malarial meds. Prevalence of the infection was also estimated before and after intervention. Data for this study were collected as part of a larger trial of 19,200 children in Burkina Faso and Mali. This secondary analysis of data from that study parallels a prior study by the lead author on resistance to Streptococcus pneumoniae.

Comments:

• Only children from Burkina Faso were included in this study. Why were data on children from Mali excluded? 

Yes, only children from Burkina Faso were included because this study was ancillary to an RCT whose primary objective was to study the impact of azithromycin administration on the reduction of infant mortality and hospitalizations in Burkina Faso and Mali. The secondary objective was to study the impact of azithromycin administration on pneumococcus and antibiotic resistance in BF and Mali. In the context of a thesis in Burkina Faso, the data from this study (only in Burkina Faso) were used to look at the serotype circulation of pneumococcus in Burkina Faso, especially since the PCV13 vaccine had been introduced into the immunization program in the country one year before the intervention began. Having noted that the prevalence of pneumococcus was decreasing over the years, the idea was to look in parallel for staphylococcus in carriage to see if it might be behind a replacement of the carriage type. We did not have access to the samples from Mali to be able to do the research on staphylococcus as well.

Given that these data were derived from a published clinical trial which included roughly twice as many children, could the authors explain why only half the children were included? The study of resistance to S. pneumoniae included all children from both B.F. and Mali.

Due to budgetary constraints, we could not conduct the research on all the samples that were taken. We therefore considered it appropriate to do random sampling on half of the children to study the circulation of serotypes in the Burkina area. And this concerned the strains of pneumococcus isolated. On the other hand, for the search for Staphylococcus, this concerned all the children sampled in the last year of the study, i.e. 778 children.

Introduction

• It seems odd that this is not identified as a sub-study of one of the studies that is referenced (ll.53-56). Although this is mentioned in the Study Population section, I think that this information should be clearly stated in the introduction (rather than referring to the parent trial as if it were a separate study).

 Added line 74

Methods

• Why were swabs from the third year only of the study included? Again, the study on S. pneumoniae contained data from all three years. Could the authors explain why they chose to limit the data to the third year only? The answer was given to the first question. Once again, this is an ancillary study to a trial and it is after having noted the presence of staph aureus during the two previous years that we wanted to really look at the prevalence of this germ, especially after having noted the decrease in the prevalence of pneumococcus with the administration of azithromycin.

• Although several studies have already been published from this trial, I still think it would be helpful to include some details about how the children were enrolled (in a dynamically stable cohort?) over the three-year period. In other areas as well, the paper would benefit from being a little more fleshed out with details, for those who have not read the other papers. 

Completed line 84-94

Results

• ll.136-37 I found this somewhat confusing as written. A clearer description would include the numbers shown in Table 1: “A total of 385 children (193 in the Azithromycin group, and 192 in the placebo group) were swabbed in 2016 before the intervention, and 393 (192 in the Azithromycin group and 201 in the placebo group) were swabbed after the intervention.” Not all children were administered AZ, so it would be more correct if the authors referred to pre- and post-intervention, rather than pre- and post-administration of AZ. 

This is done line 146

• As the authors stated that the primary objective of their study was to examine the prevalence of azithromycin-resistant S. aureus before and after the intervention, this section should precede the section on carriage with S. aureus (although that is what is stated in the Statistical Consideration section, in the introduction it is stated that the study “has examined the impact of azithromycin administration as a preventive treatment on the prevalence of S. aureus nasopharyngeal carriage and its resistance to azithromycin”). These two statements should be consistent with one another. 

We have harmonised these statements line 130-133

• The authors compare rates of S. aureus carriage in the treatment and intervention groups both before and after the intervention. They also present rates before and after within the two groups. This is confusing, given that the first set of results are presented in Table 1 and the second set are not. Perhaps a table that clearly presented both sets of results could be substituted. The same is true in their reporting of the prevalence of arithromycin-resistant S. aureus. Beautifully detailed papers are presented in the prior article on S. pneumoniae, but the table in this paper is very basic and uninformative. 

We have split the table in two: Table I: Socio-demographic characteristics of the study population Table II : Microbiological characteristics of the study population line 152 and 163

Reviewer #2: Interesting randomized controlled trial of staph resistance post azithromycin treatment. My concerns are mostly minor and easily addressable:

1. References 5-10. Your references highlight cross-sectional or longitudinal monitoring of AMR post azithromycin MDA. There are a large number of RCTs clearly showing selection of macrolide-resistant strains of pneumococcus after azithro MDA which were not referred to. If you’re not going to review that literature, consider just citing a recent systematic review (O’Brien Lancet ID, 2019). 

Ok. Pneumonia Etiology Research for Child Health (PERCH) Study Group. Causes of severe pneumonia requiring hospital admission in children without HIV infection from Africa and Asia: the PERCH multi-country case-control study [published correction appears in Lancet. 2019 Aug 31;394(10200):736]. Lancet. 2019; 394(10200):757-779. doi:10.1016/S0140-6736(19)30721-4

2. Page 3, line 64. “To date, little work has been done … other bacterial pathogens.” A number of recent studies have looked at the gut resistome post azithromycin MDA. This is in a sense looking at AMR in all bacteria, so your sentence could be reworded. You could consider referring to them (starting w/ Doan, NEJM 2019 and 2020). 

This is done and we added 2 references:

11. Doan T, Arzika AM, Hinterwirth A, Maliki R, Abdou A et al. (2019). Gut and Nasopharyngeal Macrolide Resistance in the MORDOR Study: A Cluster-Randomized Trial in Niger. N EnglJ Med. 380(23):2271–2273.

12. Doan T, Worden L, Hinterwirth A et al. (2020). Macrolide and Non macrolide Resistance with Mass Azithromycin Distribution. N Engl J Med 2020. 383 :1941-1950

3. Line 102. While the nasopharynx may be ideal for pneumococcus, would the nares have been optimal for Staphylococcus aureus recovery? Staphylococcus carriage occurs both nasal (ideally) and nasopharyngeal. 

In our study we used the nasopharyngeal swabs that had been taken for pneumococcus to also test for staphylococcus.

4. False precision throughout the paper may mislead casual readers. Eg, Table 1. 25 of 193 is 12.95%? 13.0% or even 13% would be fine given the limited number of samples. Roughly an extra digit throughout the paper may mislead the reader. Similarly, P-values .602 could be .60.

 Corrected

5. Lines 147-150. “Carriage increased after AZ administration in both the AZ and placebo

6. groups with the increase being more marked and statistically significant in the AZ group (PR=1.46 [1.06; 2.01], p=0.020 than in the children who received placebo in whom the increase was not statistically significant (PR=1.21 [0.64; 2.30], p=0.558.” Please check the meaning of this sentence, I had difficulty parsing. The most relevant contrast would be the follow-up carriage comparing the children randomized to azithro- vs placebo. The longitudinal comparisons within an arm are not nearly as interesting, as carriage or resistance could change for any number of reasons other than your intervention.

We agree with you. The longitudinal comparisons within an arm are not relevant here. We have removed it. Line 163

7. Line 153. Consider rewording “comparable”. The CI is quite broad and includes values which many might not consider “comparable”, so better to just say unable to find a significant difference in this relatively small sample. This is done line 176

8. Lines 178-180. Consider highlighting the RCT comparison between arms rather than the longitudinal comparisons within an arm, which are not as informative.

9. Line 187. Consider rewording “significantly lower” to just “lower estimates”, as I’m not sure statistical significance could actually be tested in surveys with different methods in different countries at different times. This is done Line 210

10. Line 197-199. Semicolon could be a comma? Your conclusion, that “it is essential that antibiotic resistance is monitored …” is certainly not warranted from the results of this study. In fact, you were unable to show an increase. You might say “may still be prudent to monitor”. Or you could say that the relatively wide CIs would suggest that further studies are necessary to better assess AMR. But I think the conclusion should reflect the study’s results, not opinion. 

Correction is done Line 217 and 222

11. The Figure is completely redundant with the text. Just noting, fine to leave in if you feel it conveys information in a better way.

 We keep the figure

12. I understand that the azithromycin was the randomized intervention, but you were also giving multiple doses of the antibiotic sulfidoxine. You could consider mentioning that monitoring sulfa-resistance might also be of interest. 

Thanks for the suggestion

Reviewer #3: This is a fairly straightforward report of resistance in Staph aureus isolates from a randomized trial that assessed mass azithromycin distribution vs placebo in the setting of seasonal malaria chemoprevention. I have a few suggestions to clarify the reporting.

Line 82: I think this would be much clearer if phrased: “…children received SP/AQ for seasonal malaria chemoprevention, and were randomized to either azithromycin or placebo.” Or something like that. Because right now, it sounds like one group received SP/AQ plus azithro, and the other group got placebo. 

Everything was explained in the section Study Population and Methods line 88 to 101

Line 99: please provide some details about the random selection. The underlying trial randomized households to the treatments. Were swabs also performed on all kids from randomly selected households? Or a simple random sample? Were children all taken from a single village? How many villages? This is important because the topic is a transmissible infection, and both carriage and resistance will depend not only on the intervention but also on the interventions that siblings and neighbors get.

Detail about random selection added. Please to see in the section Study Population and Methods line (88 to 101)

Line 136: this reads as if all 778 children received both swabs. But I don’t think this is the case? Consider clarifying the wording. 

All 778 children were sampled. A total of 385 children (193 in the Azithromycin group, and 192 in the placebo group) were swabbed in 2016 before the intervention, and 393 (192 in the Azithromycin group and 201 in the placebo group) were swabbed after the intervention. (Line 151)

Line 152: the “prevalence” estimate would seem to use only those kids with Staph aureus carriage as the denominator. Is this actually a prevalence? Seems like it’s moreso a frequency among the children carrying Staph aureus? Also, it would be good to clarify for the reader what the denominator is (ie, all kids regardless of Staph aureus status, vs only those with positive Staph aureus results) 

Yes, it is indeed the positive carriage of Staph aureus as the denominator. The precision has been added. It is in this case effectively frequencies. (Line 174 and 181)

Line 155/160: if the estimates being compared are only from kids with a positive Staph aureus isolate, this means the clinical trial comparison would be conditioning on a post-randomization factor (ie whether the kid grew Staph aureus), which could induce bias. It would seem less biased to me to define resistance based on the entire random sample. At least for the RCT comparison. It is well noted. 

Thanks for the input

Line 169: I always thought that the niche for Staph aureus was the nares. And yet here the nasopharynx was sampled. It might be nice to have a few sentences in the discussion about the site of sampling and whether nasopharyngeal carriage estimates would be similar to nares carriage estimates. 

Yes, in our case the same nasopharyngeal swabs that were originally used for pneumococcus were used for staphylococcus diagnostic.

---

## [Editor Report · Decision Letter 1]

26 Aug 2021

Impact of mass administration of azithromycin as a preventive treatment on the prevalence and resistance of nasopharyngeal carriage of Staphylococcus aureus

PONE-D-20-33805R1

Dear Dr. Ouangraoua,

We’re pleased to inform you that your manuscript has been judged scientifically suitable for publication and will be formally accepted for publication once it meets all outstanding technical requirements.

Kind regards,

Ray Borrow, Ph.D., FRCPath

Academic Editor

PLOS ONE
---

## [Editor Report · Acceptance letter]

5 Oct 2021

PONE-D-20-33805R1 

Impact of mass administration of azithromycin as a preventive treatment on the prevalence and resistance of nasopharyngeal carriage of *Staphylococcus aureus*

Dear Dr. Hema-Ouangraoua:

I'm pleased to inform you that your manuscript has been deemed suitable for publication in PLOS ONE. Congratulations! Your manuscript is now with our production department. 

Kind regards, 

on behalf of

Prof. Ray Borrow 

Academic Editor

PLOS ONE